# Performance during the Glittre-ADL test between patients with and without post-tuberculosis bronchiectasis: A cross-sectional study

Cristiane Pires Motta[1], Davi Luiz Olimpio da Silva[2], Lohana Resende da Costa[2], Giselle Faria Galhardo[3], Agnaldo José Lopes [1,3] *

1 Rehabilitation Sciences Post-Graduation Programme, Augusto Motta University Centre (UNISUAM), Rio de Janeiro, Brazil, 2 Faculty of Physiotherapy, Augusto Motta University Centre (UNISUAM), Rio de Janeiro, Brazil, 3 Local Development Post-Graduation Programme, Augusto Motta University Center (UNISUAM), Rio de Janeiro, Brazil

* alopes@souunisuam.com.br

## Abstract

### Background

Post-tuberculosis bronchiectasis (PTBB) is gaining recognition as an important chronic lung disease, representing a neglected condition with a significant burden for the individual. Recently, the Glittre-ADL test (TGlittre) has been proposed for the assessment of functional capacity, which incorporates tasks of daily living demanding the upper and lower extremities. This study used TGlittre to compare patients with PTBB to patients with non-post-tuberculosis bronchiectasis (NPTBB) and evaluate the determinants of performance during TGlittre.

### Methods

This is a cross-sectional study in which 32 patients with PTBB and 29 with NPTBB underwent TGlittre. In addition, they completed Short Form-36 (SF-36), handgrip strength, quadriceps muscle strength (QMS) and pulmonary function tests (PFTs).

### Results

Both PTBB and NPTBB required much more time to perform the TGlittre compared to the predicted values, although they did not differ statistically from each other [152 (124–200) vs. 145 (117–179)% predicted, p = 0.41]. Regarding the PFTs, the PTBB participants showed significantly lower values than the NPTBB participants in forced vital capacity (FVC, 60 ± 14.5 vs. 78.2 ± 22.2% predicted, p<0.001) and total lung capacity [82 (66–95) vs. 93 (82–105)% predicted, p = 0.028]. In the PTBB group, FVC (p<0.001) and QMS (p = 0.001) were the only significant independent variables to predict TGlittre time, explaining 71% of the variability in TGlittre time. In the NPTBB group, maximal expiratory pressure (p = 0.002),

**Data Availability Statement:** Follow the link to the data repository: https://osf.io/jakqm/.

**Funding:** This research was supported by the Brazilian Council of Scientific and Technological Development (Conselho Nacional de Desenvolvimento Científico e Tecnológico - CNPq, grant number #301967/2022-9), Brazil, the Carlos Chagas Filho Foundation for Research Support of the State of Rio de Janeiro (Fundação Carlos Chagas Filho de Amparo à Pesquisa do Estado do Rio de Janeiro - FAPERJ, grant numbers #E-26/010.002124/2019, #E-26/211.187/2021, #E-26/211.104/2021, and #E-26/200.929/2022), Brazil, and the Coordination for the Improvement of Higher Education Personnel (Coordenação de Aperfeiçoamento de Pessoal de Nível Superior - CAPES, FinanceCode 001, 88881.708719/2022-01, and 88887.708718/2022-00), Brazil). The funders had no role in study design, data collection and analysis, decision to publish, or preparation of the manuscript.

**Competing interests:** The authors have declared that no competing interests exist.

residual volume/TLC (p = 0.001) and QMS (p = 0.032) were the significant independent variables for predicting TGlittre time, explaining 73% of the variability in TGlittre time.

## Conclusions

PTBB patients have lower than expected performance on TGlittre, though similar to NPTBB patients. The PTBB patients had a greater reduction in lung volume than NPTBB patients. Furthermore, the performance on TGlittre in PTBB patients is largely explained by lung volume and QMS.

## Introduction

Bronchiectasis is a lung disease of diverse etiology described as one of the most neglected diseases in respiratory medicine [1]. It is a chronic condition characterized by cough, sputum production and recurrent pulmonary exacerbations, defined radiologically by abnormal bronchial dilation [2]. The prevalence of bronchiectasis continues to increase worldwide with a significant economic burden, although published data on bronchiectasis do not reflect their global burden or provide the clinical insights needed to improve management in low- and middle-income countries (LMICs) [3]. The level of Quality of Life (QoL) impairment in patients with bronchiectasis is comparable to that observed in patients with severe chronic obstructive pulmonary disease (COPD) [4]. Despite all this, historically noncystic fibrosis bronchiectasis (NCFB) has been neglected in studies, which is reflected in the absence of licensed drugs for bronchiectasis worldwide [4].

A diverse range of conditions leads to the common pathological outcome of bronchiectasis, including previous respiratory tract infections, COPD, asthma, immunodeficiency and connective tissue diseases; however, a large proportion of cases are idiopathic, reflecting the incomplete understanding of the pathogenesis of the disease [4]. Although tuberculosis (TB) is a common cause of bronchiectasis, data on this association are scarce. In 2021, approximately 11 million people became ill, and 1.6 million people died of TB worldwide, making it the second leading infectious cause of death after COVID-19, according to the World Health Organization [5]. In 2020, it was estimated that there were 155 million TB survivors worldwide [6]. In Brazil, the incidence of TB was 31.6 cases/100,000 people in 2020, placing the country among those with the highest TB burden [7]. In Brazil, almost half of the cases of bronchiectasis correspond to TB sequelae [8], unlike in European countries, where cases classified as idiopathic and non-TB postinfectious disease predominate [1].

Different levels of damage to lung function can be observed in patients with bronchiectasis. Impaired lung function usually leads to obstructive damage, although many patients may also have concomitant restrictive damage [9]. Dysfunction of mucociliary clearance, bronchial inflammation and infection, irreversible bronchial dilation and destruction of the elastic and muscular components of the bronchial walls can cause airflow limitation and restrictive damage [10]. In this population, changes in pulmonary function are associated with the etiology of the underlying disease [9]. Regarding muscle dysfunction, there are several contributors to chronic inflammatory lung diseases, including bronchiectasis. Systemic inflammation, gas exchange abnormalities, malnutrition, and medications may contribute to skeletal muscle damage [11]. Patients with bronchiectasis seem to exhibit impairment in peripheral muscle resistance and experience considerable general fatigue [10].

In patients with bronchiectasis, functional capacity usually deteriorates over time, despite appropriate clinical interventions, such as treatment with antibiotics and bronchodilators [12]. These patients commonly present progressive limitations to exercise and activities of daily living (ADLs). The main contributors to exercise intolerance seem to be associated with changes in lung mechanics, inefficiency in gas exchange, loss of peripheral muscle mass and abnormalities in cardiovascular function [13, 14]. A significant proportion of patients with NCFB show poor performance during the 6-min walk test (6MWT), which does not seem to be associated with QoL [15]. Despite being a widely used field test due to its ease of performance and low cost, the 6MWT assesses only the ability to mobilize the lower limbs, ignoring the arms in the performance of many ADLs.

Recent evidence has shown that disability-adjusted life-years attributed to post-TB lung disease, including bronchiectasis, represent approximately 50% of the total TB burden [16]. Although these patients are considered cured after treatment, significant suffering and disability may remain long after the end of treatment. The impact of post-tuberculosis bronchiectasis (PTBB) on the lungs and muscles is sparsely described in the existing literature, and there is limited knowledge about its relationship with other forms of non-cystic fibrosis bronchiectasis (NCFB) [15, 17]. An interesting way of measuring the damage of PTBBs on various organ systems is to evaluate the functional capacity to exercise using a test that simulates various ADLs. In this regard, it was proposed to use the Glitter-ADL test (TGlittre) in patients with COPD, as it includes tasks such as walking, climbing stairs and performing activities using the upper extremities [18]. Although patients' NCFB appears to have a worse performance during TGlittre compared to healthy individuals [12], there are no reports in the scientific literature of the use of this test. Thus, this study aimed to evaluate the performance of PTBB patients during TGlittre by comparing them with non-post-tuberculosis bronchiectasis (NPTBB) patients and to evaluate the determinants of performance during TGlittre.

## Methods

### Patients

Between September 2022 and March 2023, a cross-sectional study was performed with consecutive patients aged ≥18 years with PTBB at the Reference Center for Bronchiectasis at the Pedro Ernesto University Hospital of the State University of Rio de Janeiro, Rio de Janeiro, Brazil. PTBB was assigned when a history or clinical-radiological diagnosis of TB was evident in the presence of CT findings of bronchiectasis in the same lung zone previously affected by TB. In the absence of a history of TB, the diagnosis was based on the clinical judgment of the physician and CT findings consistent with TB (such as upper lobe scarring, calcification, tuberculoma and/or cavity) at sites of injury associated with bronchiectasis [19, 20]. For comparative purposes, we also evaluated a group of NPTBB patients. The following exclusion criteria were used: patients with bronchiectasis due to cystic fibrosis; patients with traction bronchiectasis caused by pulmonary fibrosis; patients who had an exacerbation of the disease within the past 30 days; individuals who were on oxygen therapy; patients with COPD or asthma associated with bronchiectasis; individuals with a history or diagnosis of pleural or cardiovascular disease; individuals with a history or diagnosis of neurological disease; and patients with a history of musculoskeletal conditions that precluded the performance of TGlittre.

The protocol was approved by the Augusto Motta University Center, Rio de Janeiro, Brazil, under protocol number 5.525.954. Written informed consent and verbal consent prior to enrolment was mandatory. Anonymous personal identifiers were used for each participant.

## Quality of life

QoL was assessed using the Short Form-36 (SF-36) validated for the Brazilian Portuguese language [21]. The questionnaire, composed of 36 items grouped into eight domains, assesses the following dimensions: physical functioning; physical role limitations; bodily pain; general health perceptions; vitality; social functioning; emotional role limitations; and mental health. Each SF-36 domain has a maximum score of 100, which indicates better QoL. There is no total score for the entire SF-36.

## Severity assessment

The severity of bronchiectasis was assessed using the E-FACED score, which includes six variables (hospitalization in the last year, forced expiratory volume in 1 sec ($FEV_1$), age, colonization with *Pseudomonas aeruginosa*, radiological extent of bronchiectasis and severity of dyspnea measured by the modified Medical Research Council scale). It has a maximum score of 9 points and categorizes the severity of the disease as mild (0–3 points), moderate (4–6 points) or severe (7–9 points) [22].

## Peripheral muscle strength

Peripheral muscle strength (PMS) was evaluated by handgrip strength (HGS) and quadriceps muscle strength (QMS). HGS was measured in kgf using a handheld digital dynamometer (SH5001, Saehan Corporation, Korea). HGS was assessed with the participants seated in an armless chair, with elbow flexion of 90˚, forearms in neutral position and wrist extension of 0–30˚ [23]. Maximum strength was assessed after a 3-s sustained contraction of the dominant hand; the highest value of three trials with 1-min intervals was considered for analysis. The QMS was measured using a tensile dynamometer with a sensor capacity of 200 kgf (E-lastic 5.0, E-sporte SE, Brazil). In the QMS evaluation, the range of motion was determined at 90˚, starting from 90˚ with the knee in flexion. Maximum quadriceps isometric contraction strength was assessed after a 5-s sustained contraction of the dominant leg, and the highest value of three attempts with 1-min intervals was considered for analysis [24].

## Pulmonary function testing

The pulmonary function tests (PFTs) consisted of spirometry, whole-body plethysmography, diffusing capacity for carbon monoxide (DLco) and measurement of respiratory muscle strength. All of these exams were performed on an HDpft 3000 device (nSpire Health, Inc., Longmont, CO, USA) and followed the recommended technical acceptability criteria [25]. All parameters of the PFTs were expressed as percentages of the predicted values [26–29]. Airflow obstruction was defined as a $FEV_1$ to forced vital capacity ($FEV_1$/FVC) <70% ratio, whereas a restrictive pattern was defined as a total lung capacity (TLC) <80% predicted [9].

## Glitter-ADL test

TGlittre was performed as previously proposed by Skumlien et al. [18] It is a circuit of functional activities in a 10 m corridor to be performed by the individual in the shortest time possible, carrying a backpack. The participant, starting from the sitting position, walked a flat course. In the middle of this journey, he must go up and down two steps of a ladder. After covering the remainder of the route, the participant is faced with three objects positioned on a shelf. Then, he must move the objects that are on the highest shelf, one by one, to the lowest shelf and then to the floor. Then, the objects must be placed again on the bottom shelf and, later, on the top shelf. After that, the participant returns, taking the reverse route. The

participant must complete five laps in the shortest time possible. The TGlittre time was compared to the Brazilian predicted values [30]. In TGlittre, the longer a participant takes to perform the test, the worse his or her functional capacity.

## Statistical analysis

The normality of data distribution was assessed using the Shapiro–Wilk test and graphical analysis of the histograms. The significance level adopted was the 5% level. Statistical analysis was performed using SPSS software, version 26. Comparisons between the two groups (PTBB and NPTBB) regarding anthropometric data, comorbidities, pulmonary function, QoL, PMS and TGlittre were analyzed using Student's $t$ test for independent samples or the Mann–Whitney $U$ test for numerical data and the chi-square test or Fisher's exact test for categorical data. The relationship between the TGlittre time and the other study variables was analyzed using Spearman's rank correlation coefficient. The correlation was classified as weak, moderate, strong or very strong given Spearman r values of 0.10–0.39, 0.40–0.69, 0.70–0.89 and 0.90–1, respectively [31]. Multivariate analysis by multiple linear regression (MLR) was used to identify the independent variables that explained the variability of the logarithm used to express the TGlittre time. This analysis was applied to the data with natural logarithmic transformation (ln TGlittre time), aiming to adapt the distribution to a parametric approach.

To provide insight into the clinical significance of the results, we calculated effect sizes using rank-biserial correlations [32] in Jeffreys's Amazing Statistics Program version 0.10.2. To provide context for interpreting the null findings, a *post hoc* power analysis was performed using GPower 3.1.1 software based on the actual sample size, the differences between the two groups, and the observed correlations between the main outcome (TGlittre time) and the other studied variables.

## Results

Among the 65 eligible patients, four were excluded. Thus, the analyzed sample consisted of 61 participants (Fig 1), 32 with PTBB and 29 with NPTBB. In the latter group, most participants had idiopathic bronchiectasis (n = 20). The mean age was 57.8 ± 14.6 and 56.1 ± 16 years (p = 0.67) in the PTBB and NPTBB groups, respectively, while the mean body mass index (BMI) was 22.3 ± 3.6 and 23.2 ± 4.2 kg/m$^2$ (p = 0.38) in the PTBB and NPTBB groups, respectively. None of the participants regularly used inhaled corticosteroids and/or long-acting bronchodilators.

Regarding PFTs, PTBB participants showed statistically lower values compared to the NPTBB group in FVC (60 ± 14.5 vs. 78.2 ± 22.2% predicted, p<0.001), FEV$_1$ (49.2 ± 17 vs. 69 ± 27.8% predicted, p = 0.001) and TLC [82 (66–95) vs. 93 (82–105)% predicted, p = 0.028]. The PTBB group showed obstruction, restriction, mixed damage and normal function in 13 (40.6%), 5 (15.6%), 11 (34.4%) and 3 (9.4%) participants, respectively, while the NPTBB group showed obstruction, restriction, mixed damage and normal function in 13 (44.8%), 1 (3.4%), 5 (17.2%) and 10 (34.5%) participants, respectively. Although the PTBB group showed a greater number of participants with severe disease according to the E-FACED score, no significant differences were found in relation to the NPTBB group; notably, there is a tendency for the PTBB group to present worse E-FACED than the NPTBB group (p = 0.054). Despite showing low SF-36 scores, PTBB and NPTBB patients did not differ statistically in the eight domains. Comparisons between the two groups regarding anthropometric data, comorbidities, pulmonary function and QoL are shown in Table 1.

In the measurement of PMS, both HGS [19 (12–24) vs. 22 (18–25) kgf, p = 0.10] and QMS [20 (13–26) vs. 24 (16–28) kgf, p = 0.23] were lower in PTBB participants than in NPTBB

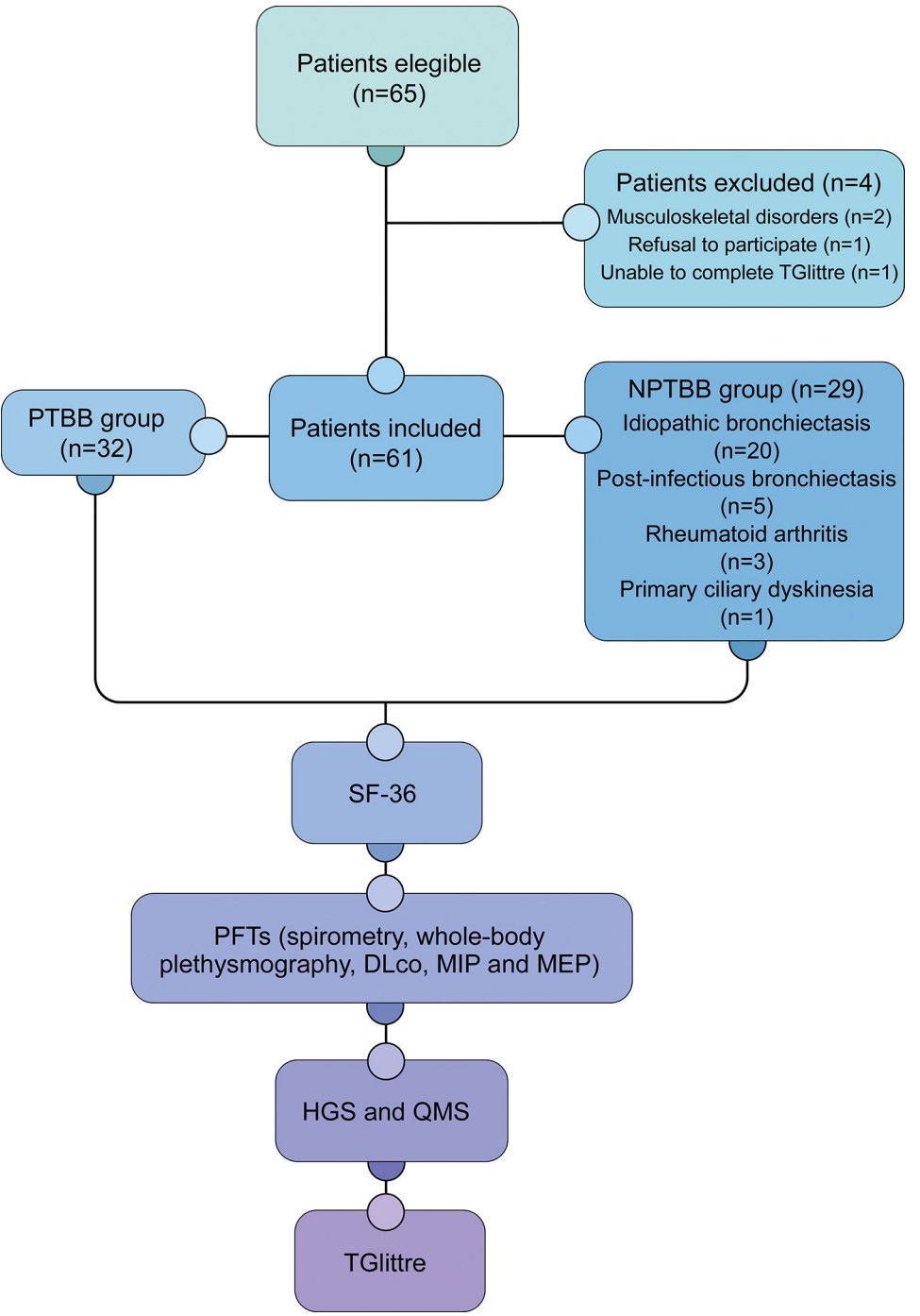

**Fig 1. Flowchart showing the study design and sample characterization.** PTBB = post-tuberculosis bronchiectasis; NPTBB = non-post-tuberculosis bronchiectasis; SF-36 = Short Form-36 (SF-36); PFTs = pulmonary function tests; DLco = diffusing capacity for carbon monoxide; MIP = maximal inspiratory pressure; MEP = maximal expiratory pressure; HGS = handgrip strength; QMS = quadriceps muscle strength; TGlittre = Glittre-ADL test.

participants, although without statistically significant differences. In TGlittre, both the PTBB and NPTBB groups required much more time to perform the tasks of TGlittre when compared to the predicted values by Reis et al. [30]. Although the PTBB group had a higher median total

**Table 1. Comparisons of anthropometric parameters, clinical data, pulmonary function, and quality of life between patients with and without post-tuberculosis bronchiectasis.**

| Variable | Total sample (n = 61) | Patients with PTBB (n = 32) | Patients with NPTBB (n = 29) | p-value* |
|---|---|---|---|---|
| **Anthropometric data** | | | | |
| Female/male | 45/16 | 23/9 | 22/7 | 0.91 |
| Age (years) | 57 ± 15.2 | 57.8 ± 14.6 | 56.1 ± 16 | 0.67 |
| Weight (kg) | 59 ± 11.3 | 58.5 ± 11 | 59.5 ± 11.9 | 0.74 |
| Height (m) | 1.61 ± 0.09 | 1.62 ± 0.09 | 1.60 ± 0.09 | 0.46 |
| BMI (kg/m$^2$) | 22.7 ± 3.9 | 22.3 ± 3.6 | 23.2 ± 4.2 | 0.38 |
| **Comorbidities, n (%)** | | | | |
| Hypertension | 12 (19.7) | 9 (28.1) | 3 (10.3) | 0.08 |
| Diabetes | 5 (8.2) | 2 (6.3) | 3 (10.3) | 0.45 |
| Cardiopathy | 3 (4.9) | 2 (6.3) | 1 (3.4) | 0.53 |
| **Pulmonary function** | | | | |
| FVC (% predicted) | 68.7 ± 20.6 | 60 ± 14.5 | 78.2 ± 22.2 | **<0.001** |
| FEV$_1$ (% predicted) | 58.6 ± 24.7 | 49.2 ± 17 | 69 ± 27.8 | **0.001** |
| FEV$_1$/FVC (%) | 67 ± 14.6 | 65.4 ± 13.7 | 68.8 ± 15.6 | 0.37 |
| DLco (% predicted) | 71 ± 30 | 64.5 ± 31.7 | 78.1 ± 26.6 | 0.075 |
| MIP (% predicted) | 57.9 ± 24.4 | 56 ± 19.4 | 60 ± 29.2 | 0.54 |
| MEP (% predicted) | 46.2 ± 20.1 | 38.8 ± 13.9 | 46.8 ± 24.8 | 0.14 |
| TLC (% predicted) | 88 (73–99) | 82 (66–95) | 93 (82–105) | **0.028** |
| RV (% predicted) | 116 (83–130) | 120 (73–132) | 108 (87–126) | 0.96 |
| RV/TLC (%) | 46.3 ± 13.8 | 49.5 ± 14.8 | 42.8 ± 11.9 | 0.060 |
| **Short Form-36** | | | | |
| Physical functioning (points) | 55 (23–80) | 65 (16–88) | 45 (25–78) | 0.50 |
| Physical role limitations (points) | 50 (0–100) | 50 (0–100) | 50 (13–100) | 0.86 |
| Bodily pain (points) | 61 (36–100) | 67 (43–100) | 41 (31–100) | 0.50 |
| General health perceptions (points) | 52 (32–67) | 52 (33–71) | 52 (30–67) | 0.63 |
| Vitality (points) | 55 (30–85) | 63 (31–85) | 55 (30–80) | 0.72 |
| Social functioning (points) | 63 (31–100) | 75 (28–100) | 50 (31–75) | 0.14 |
| Emotional role limitations (points) | 67 (0–100) | 50 (0–100) | 67 (0–100) | 0.78 |
| Mental health (points) | 64 (46–86) | 76 (45–95) | 64 (46–80) | 0.22 |
| **E-FACED** | | | | |
| Mild disease | 21 (34.4%) | 7 (21.9%) | 14 (48.3%) | 0.054 |
| Moderate disease | 25 (41%) | 14 (43.7%) | 11 (37.9%) | |
| Severe disease | 15 (24.6%) | 11 (34.4%) | 4 (13.8%) | |

Data are given as mean ± SD, median (interquartile range) or number (%). The values in bold refer to significant differences.

* Difference between PTBB and NPTBB groups. PTBB = post-tuberculosis bronchiectasis; NPTBB = non-post-tuberculosis bronchiectasis; BMI = body mass index; FVC = forced vital capacity; FEV$_1$ = forced expiratory volume in one second; DLco = diffusing capacity for carbon monoxide; MIP = maximal inspiratory pressure; MEP = maximal expiratory pressure; TLC = total lung capacity; RV = residual volume.

time than the NPTBB group, they did not differ statistically [152 (124–200) vs. 145 (117–179) % predicted, p = 0.41]. In both the PTBB and NPTBB groups, the greatest difficulty in completing the TGlittre was squatting to perform shelf tasks, which was reported by 13 (40.6%) and 9 (31%) participants, respectively. Comparisons between the two groups regarding the PMS data and TGlittre performance are shown in Table 2.

The correlations between TGlittre time and anthropometric data, pulmonary function, PMS and QoL are shown in Table 3 and Fig 2. In the total sample, TGlittre time showed strong correlations with QMS ($r_s$ = -0.75, p<0.001) and maximal expiratory pressure (MEP, $r_s$ =

**Table 2. Comparison of measures of peripheral muscle strength and performance on the Glittre-ADL test between patients with and without post-tuberculosis bronchiectasis.**

| Variable | Total sample (n = 61) | Patients with PTBB (n = 32) | Patients with NPTBB (n = 29) | p-value* |
|---|---|---|---|---|
| **Peripheral muscle strength** | | | | |
| HGS (kgf) | 20 (15–25) | 19 (12–24) | 22 (18–25) | 0.10 |
| QMS (kgf) | 22 (14–27) | 20 (13–26) | 24 (16–28) | 0.23 |
| **Glittre-ADL test** | | | | |
| Total time (min) | 4.3 (3.5–5.5) | 4.3 (3.5–5.8) | 4 (4–5) | 0.83 |
| Total time (% predicted) | 148 (121–179) | 152 (124–200) | 145 (117–179) | 0.41 |
| Highest-difficulty task, n (%) | | | | |
| Squatting to perform shelving tasks | 22 (36.1) | 13 (40.6) | 9 (31) | 0.24 |
| No difficulty | 20 (32.8) | 8 (25) | 12 (41.4) | |
| Stair tasks | 12 (19.7) | 5 (15.6) | 7 (24.1) | |
| Manual tasks | 6 (9.8) | 5 (15.6) | 1 (3.4) | |
| Chair tasks | 1 (1.6) | 1 (3.1) | 0 (0) | |

Data are given as median (interquartile range) or number (%).

* Difference between PTBB and NPTBB groups. PTBB = post-tuberculosis bronchiectasis; NPTBB = non-post-tuberculosis bronchiectasis; HGS = handgrip strength; QMS = quadriceps muscle strength.

-0.75, p<0.001) and moderate correlations with FVC ($r_s$ = -0.66, p<0.001) and maximal inspiratory pressure (MIP, $r_s$ = -0.66, p<0.001). In the PTBB group, TGlittre time showed strong correlations with FVC ($r_s$ = -0.82, p<0.001), QMS ($r_s$ = -0.78, p<0.001) and MEP ($r_s$ = -0.71, p<0.001) and moderate correlations with HGS ($r_s$ = -0.68, p<0.001) and $FEV_1$ ($r_s$ = -0.66, p<0.001). In the NPTBB group, the TGlittre time correlated strongly with MEP ($r_s$ = -0.80, p<0.001), MIP ($r_s$ = -0.75, p<0.001) and QMS ($r_s$ = -0.73, p<0.001) and moderately with residual volume-RV/TLC ($r_s$ = 0.62, p<0.001) and HGS ($r_s$ = -0.53, p<0.001).

Using the MLR in the total sample, QMS (p<0.001), FVC (p = 0.007), MEP (p = 0.023) and DLco (p = 0.028) were significant independent variables for predicting TGlittre time, largely explaining the variability of TGlittre time ($R^2$ = 0.72). In the PTBB group, FVC (p<0.001) and QMS (p = 0.001) were the only significant independent variables to predict TGlittre time, largely explaining the variability of TGlittre time ($R^2$ = 0.71). In the NPTBB group, MEP (p = 0.002), RV/TLC (p = 0.001) and QMS (p = 0.032) were the significant independent variables to predict TGlittre time, largely explaining the variability of TGlittre time ($R^2$ = 0.73).

Based on an *a priori* type-I error α = 0.05 (two-tailed), the power analysis detected significant effects in the comparisons between the two groups as follows: FVC = 97.5%; $FEV_1$ = 97%; and TLC = 91%. For the correlations with TGlittre time (total sample), significant effects were observed as follows: QMS = 97%; MEP = 97%; MIP = 94%, and FVC = 94%. These results show the adequacy of the studied sample size to obtain significant results [32].

## Discussion

In recent years, attempts have been made to establish clinical standards as a response to the questions that national TB programs have posed about post-TB disease, including the need for a submaximal field test to assess QoL and pulmonary rehabilitation [33]. In this scenario, we sought to evaluate the contribution of TGliittre in the evaluation of patients with one of the main structural sequelae of post-TB disease in an LMIC, which is bronchiectasis. We observed that PTBB and NPTBB patients have lower than expected performance on TGlittre, although they did not differ significantly from each other. Compared to the NPTBB group, the PTBB

**Table 3. Spearman's correlation coefficients for Glittre-ADL test, anthropometry data, pulmonary function, peripheral muscle strength, and quality of life.**

| Variable | Total time (% predicted) | | | | | |
|---|---|---|---|---|---|---|
| | Total sample (n = 61) | | Patients with PTBB (n = 32) | | Patients with NPTBB (n = 29) | |
| | $r_s$ | p-value | $r_s$ | p-value | $r_s$ | p-value |
| Age | 0.16 | 0.21 | 0.13 | 0.49 | 0.23 | 0.23 |
| Weight | -0.24 | 0.061 | -0.12 | 0.51 | **-0.41** | **0.029** |
| Height | -0.17 | 0.20 | -0.30 | 0.10 | -0.01 | 0.61 |
| BMI | -0.17 | 0.18 | -0.03 | 0.86 | -0.31 | 0.10 |
| FVC | **-0.66** | **<0.001** | **-0.82** | **<0.001** | **-0.52** | **0.003** |
| FEV$_1$ | **-0.58** | **<0.001** | **-0.66** | **<0.001** | **-0.44** | **0.017** |
| FEV$_1$/FVC | -0.25 | 0.056 | -0.28 | 0.12 | -0.19 | 0.34 |
| DLco | **-0.53** | **<0.001** | **-0.58** | **<0.001** | **-0.50** | **0.006** |
| MIP | **-0.66** | **<0.001** | **-0.53** | **0.002** | **-0.75** | **<0.001** |
| MEP | **-0.75** | **<0.001** | **-0.71** | **<0.001** | **-0.80** | **<0.001** |
| TLC | **-0.27** | **0.035** | -0.25 | 0.16 | -0.26 | 0.18 |
| RV | 0.21 | 0.10 | 0.19 | 0.30 | 0.27 | 0.16 |
| RV/TLC | **0.61** | **<0.001** | **0.59** | **<0.001** | **0.62** | **<0.001** |
| HGS | **-0.61** | **<0.001** | **-0.68** | **<0.001** | **-0.53** | **0.003** |
| QMS | **-0.75** | **<0.001** | **-0.78** | **<0.001** | **-0.73** | **<0.001** |
| Physical functioning | **-0.40** | **0.001** | **-0.50** | **0.004** | -0.36 | 0.052 |
| Physical role limitations | **-0.32** | **0.012** | **-0.46** | **0.008** | -0.19 | 0.33 |
| Bodily pain | -0.18 | 0.16 | -0.27 | 0.13 | -0.09 | 0.64 |
| General health perceptions | -0.21 | 0.11 | -0.26 | 0.15 | -0.14 | 0.47 |
| Vitality | -0.20 | 0.12 | -0.33 | 0.061 | -0.04 | 0.83 |
| Social functioning | -0.24 | 0.006 | **-0.50** | **0.003** | 0.02 | 0.92 |
| Emotional role limitations | -0.20 | 0.13 | -0.44 | **0.011** | 0.10 | 0.59 |
| Mental health | -0.23 | 0.076 | -0.27 | 0.13 | -0.12 | 0.52 |

PTBB = post-tuberculosis bronchiectasis; NPTBB = non-post-tuberculosis bronchiectasis; BMI = body mass index; FVC = forced vital capacity; FEV$_1$ = forced expiratory volume in one second; DLco = diffusing capacity for carbon monoxide; MIP = maximal inspiratory pressure; MEP = maximal expiratory pressure; TLC = total lung capacity; RV = residual volume; HGS = handgrip strength; QMS = quadriceps muscle strength.

group showed greater deterioration in lung function, especially in lung volume. In these patients, the worse the lung function, muscle function and QoL were, the longer the TGlittre time (meaning worse performance). In addition, lung volume and QMS are the determinants of TGlittre performance in PTBB patients, whereas in NPTBB patients, the determinants are MEP, degree of air trapping and QMS. To the best of our knowledge, this is the first study to evaluate the determinants of TGlittre in PTBB patients and NPTBB patients.

Several factors are implicated in the lower functional capacity during exercise of individuals with bronchiectasis, including increased mucus production, deterioration of respiratory function and decreased respiratory and peripheral muscle strength [10, 34]. In the present study, we observed that PTBB patients took 52% longer to perform the TGlittre tasks in relation to the Brazilian predicted values for healthy adults, although they did not differ significantly from patients with NPTBB. Using the 6MWT in a sample of NCFB patients, one-third of whom had post-TB disease, Jacques et al. [15] observed that a shorter distance walked was associated with younger age at diagnosis of bronchiectasis, lower BMI, lower FEV$_1$ and lower MEP, although they did not measure QMS. More recently, Hena et al. [12] used TGlittre in a heterogeneous group of NCFB patients that excluded PTBB patients; these authors found that NCFB individuals took 2 min longer to complete the test than healthy individuals. Interestingly, the mean

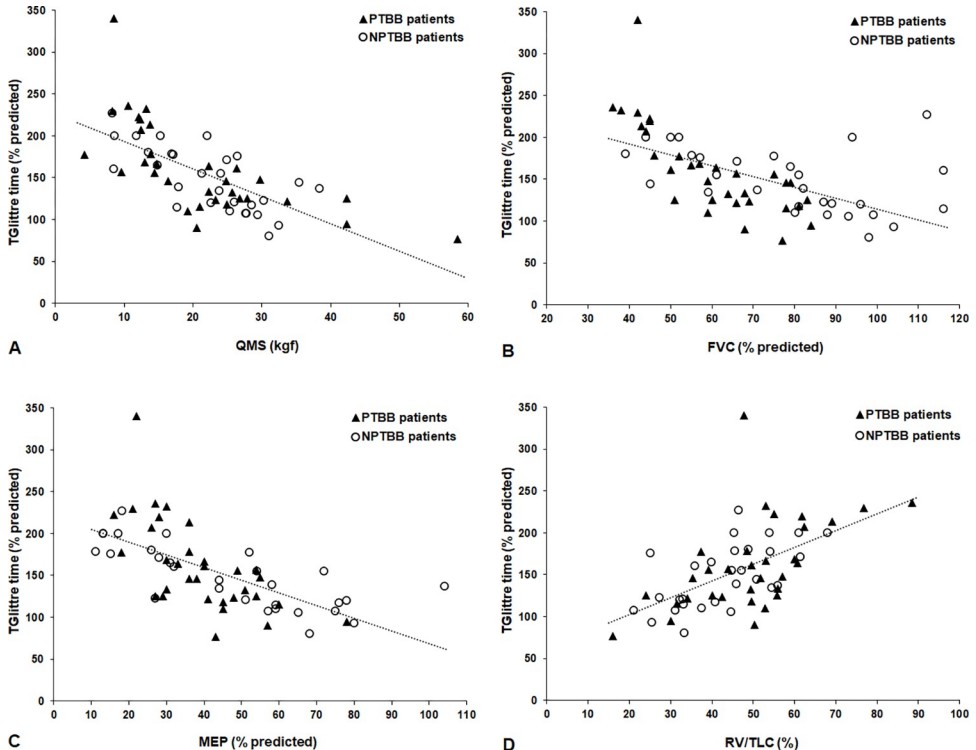

**Fig 2.** Relationships of TGlittre time with QMS ($r_s$ = -0.75, p<0.001) (A), FVC ($r_s$ = -0.66, p<0.001) (B), MEP ($r_s$ = -0.75, p<0.001) (C), and RV/TLC ($r_s$ = -0.61, p<0.001) (D). TGlittre = Glittre-ADL test; QMS = quadriceps muscle strength; FVC = forced vital capacity; MEP = maximal expiratory pressure; RV/TLC = residual volume / total lung capacity ratio; PTBB = post-tuberculosis bronchiectasis; NPTBB = non-post-tuberculosis bronchiectasis.

time of their patients to complete TGlittre (4.78 ± 1.33 min) was well above that observed in our study, which can be attributed, at least in part, to methodological differences to perform TGlittre, including permission to rest during the test; moreover, their patients had very low values of $FEV_1$ (mean 1.35 ± 0.68 L) and BMI (mean 20.1 ± 4.03 kg/m$^2$), parameters that explain part of the variability of TGlittre [30]. Notably, the greatest difficulty reported by both groups in our study was squatting during the TGlittre, which is a complex task both in terms of biomechanics and neuromuscular demands. In addition to the fact that information on sub-maximal exercise performance in patients with PTBB is scarce in the literature, we believe that TGlittre may be a useful tool in the routine evaluation of these patients and in the better assessment of their needs for therapeutic interventions such as rehabilitative strategies (especially among individuals with more severe post-TB disease).

Patients with post-TB disease are twice as likely to have abnormalities in the PFTs than the general population, and approximately 10% of them may have lost more than half of their lung function, which may have a profound impact on functional capacity over time [35]. In our sample of NCFB patients, obstructive damage was the most common finding and can be explained by several mechanisms, such as mucosal edema, presence of secretions, collapse of the airways due to the greater malleability of the affected bronchi, and concomitant infections. In a systematic review to analyze the burden of pulmonary impairment after TB as measured by PFTs, Ivanova et al. [36] showed that at least 10–15% of TB survivors had severe pulmonary impairment, much of which is attributed to bronchiectasis and subsequent obstructive damage. Importantly, we observed that, compared to the NPTBB group, PTBB patients had

statistically lower values of FVC, FEV$_1$ and TLC with restrictive damage, which is in agreement with the studies by Bogossian et al. [8], Lopes et al. [9] and Choi et al. [37], who observed lower lung volumes in PTBB patients compared to NPTBB patients. The greater reduction in lung volumes in individuals with PTBB may be due to the greater severity and intensity of bronchial and parenchymal damage. Furthermore, patients with PTBB almost always show lesions in the lung parenchyma with varying degrees of atelectasis, pulmonary fibrosis and pneumonitis, which can negatively impact lung function. Interestingly, FVC was the main predictor of TGlittre time in our subpopulation of PTBB patients, which reinforces the need for continued monitoring of lung function after discharge from TB treatment.

Peripheral muscle strength is a strong determinant of functional capacity to exercise in chronic lung diseases, including bronchiectasis [38, 39]. In people with bronchiectasis, abnormalities in the skeletal muscles (respiratory and limb) caused by inflammation, changes in gas exchange, electrolyte imbalance, inactivity and malnutrition can negatively affect functional capacity during exercise and the perception of fatigue [10, 40]. In our total sample, QMS was the variable that most explained the longer time to perform the TGlittre tasks ($p < 0.001$). Importantly, QMS was the only independent variable that entered both MLR models (PTBB and NPTBB groups) to predict TGlittre time. In this sense, we believe that the absence of significant differences in QMS between the PTBB and NPTBB patients largely explains the absence of significant differences in TGlittre time between the two groups. Evaluating the 6MWT in patients with NCFB, Ozalp et al. [10] observed that the distance traveled in the 6MWT (6MWD) was significantly related to peripheral muscle strength and endurance. These authors showed that in patients with bronchiectasis, QMS tended to be lower compared to healthy controls. Thus, muscle weakness should be a therapeutic target in patients with bronchiectasis [41], as the increase in QMS concomitantly improves the functional capacity to exercise following rehabilitation [42]. Notably, in our study, MEP was an independent variable to predict the time of TGlittre in the NPTBB group. In fact, the deterioration of respiratory muscle strength and general fatigue are findings described in patients with bronchiectasis; moreover, respiratory muscle weakness seems to be especially evident in the expiratory muscles of this patient population [10].

Although curable, TB continues to negatively affect the QoL and functioning of patients even after cure, with persistent symptoms, economic losses and impaired social life [43]. Although we observed low values for some domains of the SF-36 and some moderate correlations of these domains with TGlittre time in patients with PTBB, we did not observe significant differences between the PTBB and NPTBB groups; moreover, QoL was not a determinant in MLR for poor performance during TGlittre. In agreement with our findings, Jacques et al. [15] did not observe any difference between the SF-36 scores when the NCFB patients were separated into those with lower and higher 6MWD. Evaluating post-TB patients, including bronchiectasis patients, Daniels et al. [44] observed that QoL presented higher self-reported physical scores than mental scores. These investigators observed that the degree of impairment of pulmonary function did not show a strong association with the patients' perception of their health or capacity for effort assessed using the 6MWT. This demonstrates that the patient's perception of health may be worse than the observed physical parameters, which may negatively impact their QoL but not their functional capacity during exercise.

The strength of this study is that it evaluated the impact of pulmonary and extrapulmonary contributors on TGlittre in patients with PTBB compared to a group of patients with NPTBB. However, our study has some limitations that should be noted. First, our sample is relatively small and from a single center. Second, the study has a cross-sectional design that does not allow the examination of the temporal relationship between TGlittre and the other variables. Third, the NPTBB group was composed of several different clinical conditions, although our

main interest was to evaluate the impact of bronchiectasis in post-TB disease on TGlittre. In addition, we could have used more specific questionnaires for patients with bronchiectasis, although the SF-36 showed good internal consistency in all its domains in this population [45]. Despite these limitations, our findings may contribute to randomized controlled trials in patients with bronchiectasis associated with post-TB disease, as it is a neglected condition that deteriorates lung function, impairs QoL, reduces functional capacity and has important implications for economic costs, especially for LMICs. Furthermore, the results of the present study may provide information for outcome measures in pulmonary rehabilitation programs in patients with PTBB.

In conclusion, our study shows that PTBB patients have lower than expected performance on TGlittre, though similar to NPTBB patients. These patients had a greater reduction in lung volume than NPTBB patients. There are relationships between TGlittre time and pulmonary function, muscle function and QoL. The performance on TGlittre in PTBB patients is largely explained by lung volume and QMS, while in NPTBB patients, it is largely explained by respiratory muscle strength, degree of air trapping and QMS.

## Supporting information

**S1 Checklist. STROBE statement—Checklist of items that should be included in reports of observational studies.**
(PDF)

## Author Contributions

**Conceptualization:** Cristiane Pires Motta, Davi Luiz Olimpio da Silva, Lohana Resende da Costa, Giselle Faria Galhardo, Agnaldo José Lopes.

**Data curation:** Cristiane Pires Motta, Agnaldo José Lopes.

**Formal analysis:** Cristiane Pires Motta, Davi Luiz Olimpio da Silva, Lohana Resende da Costa, Agnaldo José Lopes.

**Funding acquisition:** Agnaldo José Lopes.

**Investigation:** Cristiane Pires Motta, Davi Luiz Olimpio da Silva, Lohana Resende da Costa, Agnaldo José Lopes.

**Methodology:** Cristiane Pires Motta, Davi Luiz Olimpio da Silva, Lohana Resende da Costa, Giselle Faria Galhardo, Agnaldo José Lopes.

**Project administration:** Cristiane Pires Motta.

**Supervision:** Agnaldo José Lopes.

**Validation:** Cristiane Pires Motta, Davi Luiz Olimpio da Silva, Lohana Resende da Costa, Giselle Faria Galhardo.

**Writing – original draft:** Cristiane Pires Motta, Davi Luiz Olimpio da Silva, Lohana Resende da Costa, Giselle Faria Galhardo, Agnaldo José Lopes.

**Writing – review & editing:** Cristiane Pires Motta, Davi Luiz Olimpio da Silva, Lohana Resende da Costa, Giselle Faria Galhardo, Agnaldo José Lopes.

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
