## [Decision Letter · Decision Letter 0]

31 Jul 2023

PONE-D-23-17193Performance during the Glittre-ADL test: A comparative study between patients with and without posttuberculosis bronchiectasisPLOS ONE

Dear Dr. Lopes,

Thank you for submitting your manuscript to PLOS ONE. After careful consideration, we feel that it has merit but does not fully meet PLOS ONE’s publication criteria as it currently stands. Therefore, we invite you to submit a revised version of the manuscript that addresses the points raised during the review process.

Please find the attached reviewer comments and requesting you to amend the changes as suggested. 

We look forward to receiving your revised manuscript.

Kind regards,

Sampath Kumar Amaravadi, Ph.D

Academic Editor

PLOS ONE

Additional Editor Comments:

Dear Author,

Please find the the reviewer comments attached and requesting you to amend the changes suggested.

Reviewers' comments:

Reviewer's Responses to Questions

**Comments to the Author**

1. Is the manuscript technically sound, and do the data support the conclusions?

Reviewer #1: Partly

Reviewer #2: Yes

2. Has the statistical analysis been performed appropriately and rigorously? 

Reviewer #1: Yes

Reviewer #2: Yes

3. Have the authors made all data underlying the findings in their manuscript fully available?

Reviewer #1: Yes

Reviewer #2: Yes

4. Is the manuscript presented in an intelligible fashion and written in standard English?

Reviewer #1: Yes

Reviewer #2: Yes

5. Review Comments to the Author

Reviewer #1: This is a cross-sectional study that proposed to evaluate the performance of Post-tuberculosis bronchiectasis patients during TGlittre by comparing them with nonposttuberculosis bronchiectasis patients and to evaluate the determinants of performance during TGlittre. In addition, this patients completed Short Form-36, handgrip strength, quadriceps muscle strength and pulmonary function tests.

The article presents some corrections and adjustments to be made, in addition to some doubts about the study methods.

The comments are presented in an attached text.

Reviewer #2: Dear Editor,

This well-designed cross-sectional study provides valuable insights into the functional capacity of patients with posttuberculosis bronchiectasis and non-posttuberculosis bronchiectasis using the TGlittre test. The study lays a foundation for developing effective interventions and personalized care for individuals burdened by these chronic lung diseases by unravelling key determinants of functional performance. The study's significance lies in its contributions to bronchiectasis research and its potential to impact clinical practice and improve patients' quality of life. I have minor comments.

Regards,

Fernando Guimaraes, PT, PhD

6. PLOS authors have the option to publish the peer review history of their article (what does this mean?). If published, this will include your full peer review and any attached files.

Reviewer #1: No

Reviewer #2: No

---

## [Author Response · Author response to Decision Letter 0]

8 Aug 2023

REVIEWER #1:

First, we would like to thank you for your time and comments, which have indeed helped us improve the manuscript. We agree that some points in the initial version of the manuscript should have been described in more detail. We have replied to each of your comments below. The modifications made to the text are highlighted.

This is a cross-sectional study that proposed to evaluate the performance of Post-tuberculosis bronchiectasis patients during TGlittre by comparing them with nonposttuberculosis bronchiectasis patients and to evaluate the determinants of performance during TGlittre. In addition, this patients completed Short Form-36, handgrip strength, quadriceps muscle strength and pulmonary function tests. The article presents some corrections and adjustments to be made, in addition to some doubts about the study methods. The comments are presented in an attached text.

AUTHORS:

Thank you for your comments.

Title:

- As followed by the authors, STROBE suggests that the type of study be included in the title.

AUTHORS:

As required, the title of the manuscript has been modified as follows: “Performance during the Glittre-ADL test between patients with and without post-tuberculosis bronchiectasis: A cross-sectional study.”

Abstract:

- Check the correct spelling of Post Tuberculosis.

AUTHORS:

Thank you for your observation. We adopted the spelling "post-tuberculosis" throughout the manuscript.

- “and evaluate the determinants of lung performance during TGlittre.”- In the text of the article, lung performance is not mentioned. Please change the abstract.

AUTHORS:

We apologize for the typo. The sentence was corrected as follows: "... and evaluate the determinants of performance during TGlittre."

- Conclusion: see comments on the conclusion in the text.

AUTHORS:

As required, the first sentence of the Abstract conclusion has been amended as follows: “PTBB patients have lower than expected performance on TGlittre, though similar to NPTBB patients. The PTBB patients had a greater reduction in lung volume than NPTBB patients. Furthermore, the performance on TGlittre in PTBB patients is largely explained by lung volume and QMS.”

Introduction:

- “In Brazil, the incidence of TB was 31.6 cases/100,000 people in 2020, placing the country among those with the highest TB burden [7].” – Authors should review this citation. This information is from another article cited in reference article 7. This data is presented in the introduction of reference 7, following another reference.

AUTHORS:

Thank you for your observation. The citation has been corrected as per the reference [7].

[7] Secretaria de Vigilância em Saúde. Ministério da Saúde. 2021 Mar [cited 31 July 2023]. In: Boletim Epidemiológico Tuberculose 2021 [Internet]. Brasília: Brazil. Available from: https://www.gov.br/saude/pt-br/centrais-de-conteudo/publicacoes/boletins/epidemiologicos/especiais/2021/boletim-tuberculose-2021_24.03.

- “...this test has never been used previously in PTBB patients.”- I suggest that the authors change this sentence to “there are no reports in the scientific literature of the use of this test”.

AUTHORS:

As required, the sentence has been modified accordingly.

Methods:

- How did the authors calculate the sample size? I also suggest that the authors present a flowchart with the study design and sample characterization.

AUTHORS:

As you suggested, we made a flowchart with the study design and sample characterization. We performed a power analysis to determine the power of the results, especially for significant results. Thus, the following sentences were added to the manuscript:

• Statistical analysis: “To provide insight into the clinical significance of the results, we calculated effect sizes using rank-biserial correlations [32] in Jeffreys’s Amazing Statistics Program version 0.10.2. To provide context for interpreting the null findings, a post hoc power analysis was performed using GPower 3.1.1 software based on the actual sample size, the differences between the two groups, and the observed correlations between the main outcome (TGlittre time) and the other studied variables.”

• Results: “Based on an a priori type-I error α = 0.05 (two-tailed), the power analysis detected significant effects in the comparisons between the two groups as follows: FVC = 97.5%; FEV1 = 97%; and TLC = 91%. For the correlations with TGlittre time (total sample), significant effects were observed as follows: QMS = 97%; MEP = 97%; MIP = 94%, and FVC = 94%. These results show the adequacy of the studied sample size to obtain significant results [32].”

- How was the recruitment of these patients?

AUTHORS:

Thank you for your observation. Our institution is one of the reference centers for bronchiectasis in Brazil, where patients are meticulously investigated as to the cause of bronchiectasis and regularly followed up. With that in mind, the following sentence was added to the manuscript: “Between September 2022 and March 2023, a cross-sectional study was performed with consecutive patients aged ≥18 years with PTBB at the Reference Center for Bronchiectasis at the Pedro Ernesto University Hospital of the State University of Rio de Janeiro, Rio de Janeiro, Brazil.”

- I suggest making explicit how the clinical diagnosis of bronchiectasis is performed.

AUTHORS:

As required, we explain how the clinical diagnosis of bronchiectasis was made as follows: “PTBB was assigned when a history or clinical-radiological diagnosis of TB was evident in the presence of CT findings of bronchiectasis in the same lung zone previously affected by TB. In the absence of a history of TB, the diagnosis was based on the clinical judgment of the physician and CT findings consistent with TB (such as upper lobe scarring, calcification, tuberculoma and/or cavity) at sites of injury associated with bronchiectasis [19,20].”

- “or clinical evidence of TB”- what clinical evidence?

AUTHORS:

We apologize for the incorrectly used expression. The term "clinical evidence" was replaced by "clinical-radiological diagnosis".

- Were patients with COPD or asthma associated with bronchiectasis excluded?

AUTHORS:

Patients with COPD or asthma associated with bronchiectasis were excluded from our study, although no patient evaluated for inclusion had any of these diagnoses. Information has been added to the Methods section as follows: “... patients with COPD or asthma associated with bronchiectasis.”

- Exclusion criteria: I suggest adding individuals with a neurological history or diagnosis.

AUTHORS:

As suggested, this exclusion criteria was added to the manuscript as follows: “... individuals with a history or diagnosis of neurological disease.”

- Were the patients using any medication that would influence test performance, such as bronchodilators?

AUTHORS:

There is insufficient evidence to recommend the routine use of inhaled corticosteroids or long-acting bronchodilators in patients with bronchiectasis without a diagnosis of asthma or COPD. Since patients with asthma or COPD were excluded from our study, no patient routinely used these medications. Thus, the following sentence has been added to the manuscript: “None of the participants regularly used inhaled corticosteroids and/or long-acting bronchodilators.”

- The acronym CAAE (Protocol of Approval in the Ethics Committee) means Certificate of Presentation of Ethical Appreciation, which indicates that the project was sent to the Ethics Committee, but does not mean approval. I suggest showing the approval number.

AUTHORS:

As required, the CAAE number was replaced by the approval number as follows: “The protocol was approved by the Augusto Motta University Center, Rio de Janeiro, Brazil, under protocol number 5.525.954.”

- Did the authors use the FACED, BSI or other document to assess the severity of bronchiectasis in these patients?

AUTHORS:

Thank you for your comment. We evaluated E-FACED scores. Thus, we have added this information in Table 1, as well as the comparison between groups. Additionally, we entered the following information in the new version of the manuscript:

• Methods: “The severity of bronchiectasis was assessed using the E-FACED score, which includes six variables (hospitalization in the last year, forced expiratory volume in 1 sec (FEV1), age, colonization with Pseudomonas aeruginosa, radiological extent of bronchiectasis and severity of dyspnea measured by the modified Medical Research Council scale). It has a maximum score of 9 points and categorizes the severity of the disease as mild (0–3 points), moderate (4–6 points) or severe (7–9 points) [22].”

• Results: “Although the PTBB group showed a greater number of participants with severe disease according to the E-FACED score, no significant differences were found in relation to the NPTBB group; notably, there is a tendency for the PTBB group to present worse E-FACED than the NPTBB group (p=0.054).”

- The authors used the SF-36 to assess the quality of life in patients with bronchiectasis. Why was this questionnaire used? Are there other questionnaires that could be used, more specific for this population?

AUTHORS:

Thank you for your comments. The authors agree that we could have used more specific questionnaires for this population. However, studies on the psychometric properties of the Brazilian Portuguese versions of these more specific questionnaires have only occurred in recent years. A recent meta-analysis showed that future studies should focus on medium to long term test-retest reliability, responsiveness and minimal clinically important difference (MCID) in those HRQoL questionnaires that show potential in bronchiectasis [1]. Despite these considerations, we added the use of the SF-36 to our study as a limitation as follows: “In addition, we could have used more specific questionnaires for patients with bronchiectasis, although the SF-36 showed good internal consistency in all its domains in this population [45].”

[1] McLeese RH, Spinou A, Alfahl Z, Tsagris M, Elborn JS, Chalmers JD, et al. Psychometrics of health-related quality of life questionnaires in bronchiectasis: a systematic review and meta-analysis. The European Respiratory Journal. 2021; 58(5):2100025.

- “Maximum strength was assessed after a 5-s sustained contraction of the dominant leg, and the highest value of three attempts with 1-min intervals was considered for analysis [20].”- I suggest reviewing the reference used to perform this test. Reference 20 is from an article on functional capacity in women with rheumatoid arthritis.

AUTHORS:

Thank you for your observation. The citation has been corrected as per the reference [24].

[24] Ushiyama N, Kurobe Y, Momose K. Validity of maximal isometric knee extension strength measurements obtained via belt-stabilized hand-held dynamometry in healthy adults. Journal of Physical Therapy Science. 2017; 29(11): 1987–92. https://doi.org/10.1589/jpts.29.1987 PMID: 29200641; PubMed Central PMCID: PMC5702831.

- “the variability of the TGlitre logarithmic time”. – I kindly ask you to correct the name of the test.

AUTHORS:

As required, the sentence has been rewritten to better describe the statistical analysis used as follows: “Multivariate analysis by multiple linear regression (MLR) was used to identify the independent variables that explained the variability of the logarithm used to express the TGlittre time. This analysis was applied to the data with natural logarithmic transformation (ln TGlittre time), aiming to adapt the distribution to a parametric approach.”

Results:

- Table 2 – In this table, was any data presented in mean ± SD? If not, kindly remove it from the legend.

AUTHORS:

Thank you for the observation. The correction was made accordingly.

- In some tables, the results presented were in median and others in mean and standard deviation. Why did the authors recommend presenting most of the results in median and not in mean?

AUTHORS:

After checking the normality of data distribution using the Shapiro‒Wilk test and graphical analysis of the histograms, most variables did not show a Gaussian distribution. To follow greater statistical rigor, variables with non-normal distribution were presented as median and interquartile ranges, while variables with normal distribution were presented as mean ± standard deviation. Along the same lines, we had to use the natural logarithmic transformation (ln TGlittre time), aiming to adapt the distribution to a parametric approach for multivariate analysis.

Discussion:

- “It is estimated that years of life lost account for approximately one-quarter, whereas years lived with disability account for three-quarters (77%) of the total burden of disease associated with post-TB pulmonary impairment [16].”- I apologize, as I did not understand this sentence presented in the discussion.

AUTHORS:

Thank you for your comment. One means of measuring the global burden of disease is disability-adjusted life-years (DALYs), estimated by combining the burden of mortality (i.e., years of life lost [YLL]) and the burden of morbidity (i.e., years living with a disability [YLD]). DALYs has been used to compare the impact of diseases that cause premature death but little disability with diseases that do not cause death but lead to disability. Recent evidence has shown that DALYs attributed to post-TB lung disease (PTLD)—where bronchiectasis is one of the main sequel—represent about 50% of the total burden of TB in patients. In other words, although we classify patients as “cured” after successful treatment, significant distress and disability may remain long after treatment has ended. It has been estimated that YLL contributes approximately one-quarter, while YLD contributes three-quarters (77%) of PTLD's burden. Furthermore, PTLD accounts for 97.4% of YLDs, whereas illness before completion of treatment, including acute treatment-related side effects, accounts for the rest of YLDs.

 Although there is worldwide concern about the burden associated with PTLD, we chose to delete this first sentence from the Discussion section in order to move the discussion towards our results.

- “We observed that PTBB and NPTBB patients had poor performance on TGlittre, although they did not differ significantly from each other.”- This is the main objective of the study and the authors did not discuss it. Why do you think there was no significant difference? And clinical difference?

AUTHORS:

Thank you for the observation. We added our hypothesis about the fact that PTBB and NPTBB patients had poor performance on TGlittre, although they did not differ significantly from each other, as follows: “In our total sample, QMS was the variable that most explained the longer time to perform the TGlittre tasks (p<0.001). Importantly, QMS was the only independent variable that entered both MLR models (PTBB and NPTBB groups) to predict TGlittre time. In this sense, we believe that the absence of significant differences in QMS between the PTBB and NPTBB patients largely explains the absence of significant differences in TGlittre time between the two groups. Evaluating the 6MWT in patients with NCFB, Ozalp et al. [10] observed that the distance traveled in the 6MWT (6MWD) was significantly related to peripheral muscle strength and endurance. These authors showed that in patients with bronchiectasis, QMS tended to be lower compared to healthy controls. Thus, muscle weakness should be a therapeutic target in patients with bronchiectasis [41], as the increase in QMS concomitantly improves the functional capacity to exercise following rehabilitation [42].”

- “In these patients, the longer the TGlittre time was, the worse the lung function, muscle function and QoL.”- Wouldn't it be the other way around?

AUTHORS:

Thank you for your observation. The sentence was reworded as follows: “In these patients, the worse the lung function, muscle function and QoL were, the longer the TGlittre time (meaning worse performance).”

- “In addition, lung volume and QMS are the determinants of TGlittre performance in PTBB patients, whereas in NPTBB patients, the determinants are respiratory muscle strength, degree of air trapping and QMS”. – Respiratory muscle or expiratory muscle?

AUTHORS:

Thank you for your observation. The sentence has been modified as follows: “In addition, lung volume and QMS are the determinants of TGlittre performance in PTBB patients, whereas in NPTBB patients, the determinants are MEP, degree of air trapping and QMS.”

- “To the best of our knowledge, this is the first study to evaluate the determinants of TGlittre in PTBB patients compared to those in NPTBB patients. “-But the authors did not evaluate the determinants in comparison, they are individual determinants for that specific group.

AUTHORS:

Thank you for your observation. The sentence has been modified as follows: “To the best of our knowledge, this is the first study to evaluate the determinants of TGlittre in PTBB patients and NPTBB patients.”

- “In the present study, we observed that PTBB patients took 52% longer to perform the TGlittre tasks compared to NPTBB patients, although the groups did not differ significantly.”- It was not 52% compared to the other group, but in relation to the predicted value, described in the article by Reis et al. (ref 26), as described by the authors on page 13.

AUTHORS:

We apologize for the typo. The sentence has been reformulated as follows: “In the present study, we observed that PTBB patients took 52% longer to perform the TGlittre tasks in relation to the Brazilian predicted values for healthy adults, although they did not differ significantly from patients with NPTBB.”

- “...to methodological differences in the performance of TGlittre.”- What would those differences be?

AUTHORS:

As required, we point out some differences between the two studies as follows: “More recently, Hena et al. [12] used TGlittre in a heterogeneous group of NCFB patients that excluded PTBB patients; these authors found that NCFB individuals took 2 min longer to complete the test than healthy individuals. Interestingly, the mean time of their patients to complete TGlittre (4.78 ± 1.33 min) was well above that observed in our study, which can be attributed, at least in part, to methodological differences to perform TGlittre, including permission to rest during the test; moreover, their patients had very low values of FEV1 (mean 1.35 ± 0.68 L) and BMI (mean 20.1 ± 4.03 kg/m2), parameters that explain part of the variability of TGlittre [30].”

- “In our sample of NCFB patients, obstructive damage was the most common finding and can be explained by several mechanisms, such as mucosal edema, presence of secretions, collapse of the airways due to the greater malleability of the affected bronchi, and concomitant infections [9,31].”- I apologize, but I did not understand the reason for references 9 and 31 to be in this sentence, since the authors are commenting on their results and not comparing or discussing with the scientific literature.

AUTHORS:

We apologize for the error. The references were deleted from the sentence.

- “Peripheral muscle strength is a strong determinant of functional capacity to exercise in chronic lung diseases [34].”- Is this reference correct? The article is about elderly and hand grip only. And the mean/median age of the groups were not elderly.

AUTHORS:

We changed the reference to two more suitable and more recent ones.

Conclusion:

- “In conclusion, our study shows that PTBB patients have poor performance on TGlittre, which is similar to that of NPTBB patients.” - What would be poor performance? Wouldn't it be better below expected? The two groups were lower than expected, but it is worth mentioning the main objective of the study, informing that there was no difference between the 2 groups.

AUTHORS:

Thank you for your excellent contributions to the improvement of the manuscript. In particular, the authors fully agree with you for the modification of the first sentence of the conclusion, which has been amended as follows: “In conclusion, our study shows that PTBB patients have lower than expected performance on TGlittre, though similar to NPTBB patients. These patients had a greater reduction in lung volume than NPTBB patients. There are relationships between TGlittre time and pulmonary function, muscle function and QoL. The performance on TGlittre in PTBB patients is largely explained by lung volume and QMS, while in NPTBB patients, it is largely explained by respiratory muscle strength, degree of air trapping and QMS.”

REVIEWER #2:

First, we would like to thank you for your time and comments, which have indeed helped us improve the manuscript. We agree that some points in the initial version of the manuscript should have been described in more detail. We have replied to each of your comments below. The modifications made to the text are highlighted.

This well-designed cross-sectional study provides valuable insights into the functional capacity of patients with posttuberculosis bronchiectasis and non-posttuberculosis bronchiectasis using the TGlittre test. The study lays a foundation for developing effective interventions and personalized care for individuals burdened by these chronic lung diseases by unravelling key determinants of functional performance. The study's significance lies in its contributions to bronchiectasis research and its potential to impact clinical practice and improve patients' quality of life. I have minor comments.

AUTHORS:

Thank you for your comments.

Introduction

Replace the sentence "Quality of life (QoL) impairment in patients with bronchiectasis is equivalent to the impairment observed in patients with severe chronic obstructive pulmonary disease (COPD)" by "The level of Quality of Life (QoL) impairment in patients with bronchiectasis is comparable to that observed in patients with severe chronic obstructive pulmonary disease (COPD)."

AUTHORS:

As required, the sentence was replaced accordingly.

Replace the sentence "In the presence of bronchiectasis, peripheral muscle resistance and general fatigue seem to be significantly impaired [10]" by "Patients with bronchiectasis seem to exhibit impairment in peripheral muscle resistance and experience considerable general fatigue [10]."

AUTHORS:

As required, the sentence was replaced accordingly.

Replace the sentence "The impact of posttuberculosis bronchiectasis (PTBB) on the lungs and muscles is sparsely described in the existing literature, and there is limited knowledge about its relationship with other forms of non-cystic fibrosis bronchiectasis (NCFB) [7,15]" by "The impact of posttuberculosis bronchiectasis (PTBB) on the lungs and muscles is sparsely described in the existing literature, and there is limited knowledge about its relationship with other forms of non-cystic fibrosis bronchiectasis (NCFB) [7,15]."

AUTHORS:

As required, the sentence was replaced accordingly.

Methods

QOL: "…validated for the Brazilian Portuguese language."

AUTHORS:

As required, the sentence was rephrased accordingly.

Make it clear that each domain in the SF-36 Quality of Life (QOL) questionnaire has a maximum score of 100, and there is no total score for the entire SF-36 questionnaire.

AUTHORS:

Thank you for your suggestion. The sentences were rewritten as follows: “Each SF-36 domain has a maximum score of 100, which indicates better QoL. There is no total score for the entire SF-36.”

Was the measurement of quadriceps strength performed isometrically? If yes, please refer to it as 'maximum quadriceps isometric contraction strength'.

AUTHORS:

Thank you for the observation. The expression has been modified to "maximum quadriceps isometric contraction strength".

Check the units (kg or kgf?). In "Results" the values are reported in kgf for QMS and Handgrip.

AUTHORS:

Thank you for the observation. Values were expressed as kgf. Modifications were made accordingly.

Results

Do not repeat the results in text and tables.

AUTHORS:

Thank you for the observation. Repeats have been deleted.

Discussion/Conclusion

The sentence "Our findings may provide information for outcome measures in pulmonary rehabilitation programs in patients with PTBB" should be moved to the Discussion. Additionally, I believe that your findings have the potential to offer valuable insights not only for outcome measures but also for proposing interventions in randomized controlled trials (RCTs).

AUTHORS:

As required, the sentence was moved to the Discussion section as follows: “Despite these limitations, our findings may contribute to randomized controlled trials in patients with bronchiectasis associated with post-TB disease, as it is a neglected condition that deteriorates lung function, impairs QoL, reduces functional capacity and has important implications for economic costs, especially for LMICs. Furthermore, the results of the present study may provide information for outcome measures in pulmonary rehabilitation programs in patients with PTBB.”

---

## [Editor Report · Decision Letter 1]

17 Aug 2023

Performance during the Glittre-ADL testbetween patients with and without post-tuberculosis bronchiectasis: A cross-sectional study

PONE-D-23-17193R1

Dear Dr. Lopes,

We’re pleased to inform you that your manuscript has been judged scientifically suitable for publication and will be formally accepted for publication once it meets all outstanding technical requirements.

Kind regards,

Sampath Kumar Amaravadi, Ph.D

Academic Editor

PLOS ONE
---

## [Editor Report · Acceptance letter]

25 Aug 2023

PONE-D-23-17193R1 

Performance during the Glittre-ADL test between patients with and without post-tuberculosis bronchiectasis: A cross-sectional study 

Dear Dr. Lopes:

I'm pleased to inform you that your manuscript has been deemed suitable for publication in PLOS ONE. Congratulations! Your manuscript is now with our production department. 

Kind regards, 

on behalf of

Dr Sampath Kumar Amaravadi 

Academic Editor

PLOS ONE